# OpenReview forum: "Fairness Failure Modes of Multimodal LLMs"
_ICLR.cc/2026/Workshop/AFAA — AFAA 2026 Oral_

### Official Review · Reviewer_fCsm · 2026-02-11
**Review of "Fairness Failure Modes of Multimodal LLMs"**

**Rating:** 5
**Confidence:** 4

**Summary:**

This paper introduces MultiBBQ, a fairness evaluation benchmark for Multimodal Large Language Models (MLLMs), extending the text-only BBQ benchmark to multimodal contexts. The benchmark pairs AI-generated photorealistic images with textual contexts across four demographic categories (Gender, Race, Religion, Age) and evaluates models in three settings: Pure Visual (PV) ambiguous, Visual+Language (VL) ambiguous, and VL disambiguated. The authors propose two complementary metrics (Fairness Score and Bias Score) and address shortcut reasoning (by replacing demographic terms with neutral expressions) and data contamination (via synthetic images). Through comprehensive evaluation of 28 MLLMs (6 proprietary, 22 open-source), they identify four "Fairness Failure Modes": (1) divergent failure patterns between proprietary and open-source models, (2) fairness degradation from input/model factors, (3) bias amplification relative to backbone LLMs, and (4) limited effectiveness of existing mitigation strategies.

**Strengths:**

1. **Important, timely problem with high workshop relevance.** This paper addresses a genuinely pressing question: how do we evaluate fairness in multimodal LLMs as they are deployed in high-stakes settings? Existing text-only frameworks are insufficient for capturing bias patterns arising from visual inputs or cross-modal interactions, and this work fills that gap directly. The paper is an excellent fit for this workshop.

2. **High-quality benchmark design and release.** MultiBBQ is not only a contribution to the research paper itself but a practical asset for the broader community. The three-context design (PV ambiguous, VL ambiguous, VL disambiguated) enables precise attribution of fairness failures to specific reasoning modes. The release of the full dataset and code ensures lasting value beyond this paper.

3. **Careful experimental design at impressive scale.** The paper stands out for the rigor with which it addresses known shortcomings of prior benchmarks. The shortcut reasoning mitigation (replacing demographic terms with neutral positional expressions) is a clever and necessary design choice that prevents models from trivially inferring correct answers from text alone. Equally impressive is the control of confounding attributes: when assessing Gender bias, both persons in the image are matched on Race, Religion, and Age, isolating the target demographic. That this level of care is maintained across 28 models, four demographic categories, three context types, and extensive ablations on image quality, quantization, and temperature speaks to both the very high quality and the impressive scale of the experimental methodology.

4. **Thorough validity checking through ablations.** The paper goes well beyond standard benchmarking with systematic ablations: multiple image generators are tested, yielding near-perfect cross-generator correlation; image quality perturbations (noise, brightness, contrast, compression, resolution) are applied; quantization effects are measured; and decoding temperature sensitivity is evaluated. The transferability to real-world images is particularly convincing. This extensive validation builds strong confidence that the findings are robust and not artifacts of specific experimental choices.

5. **Compelling and insightful results.** The experiments yield a wealth of findings that are both novel and practically important. The divergence between proprietary and open-source models, where proprietary models over-refuse in disambiguated contexts while open-source models fail to abstain in ambiguous ones, is a striking and previously undocumented pattern. The finding that verbal context reduces bias (cross-modal reasoning outperforms pure visual reasoning), that image noise is the most detrimental input factor for fairness, that MLLMs amplify bias relative to their backbone LLMs, that larger models within families exhibit less bias, and that common mitigation strategies show limited or inconsistent effectiveness; these are all actionable insights that should inform both model development and deployment decisions.

6. **Clear writing and well-presented results.** The paper is clearly written throughout, with a logical flow from problem formulation to benchmark design to experimental results. The four Fairness Failure Modes are crisply articulated and easy to remember. Figures and tables are well-designed and effectively communicate the results, Table 2's color-coded heatmap enables rapid visual comparison across 28 models, and Figures 2-4 clearly convey the factor analysis and mitigation results.

7. **Reproducibility.** Released code and dataset, detailed model lists with download links, and explicit prompts in the appendix all support reproducibility.

**Weaknesses:**

Overall the paper is very strong and thorough. I recommend the authors include a formal limitations/future work section. Here are a few suggestions to mention:

1.  **Reliance on AI-generated images introduces its own biases.** While synthetic images enable controllability, the image generators (GPT Image Generation, Imagen 4 Ultra) themselves encode demographic biases and stereotypical representations. For instance, what does a "Muslim person" or "old-timer" look like according to these generators? The paper does not discuss whether the generated images themselves might introduce stereotypical visual cues that could confound the evaluation. A brief human audit of image quality beyond general correctness, specifically examining stereotypical portrayals, would strengthen the work.

2. **The "over-refusal" finding could benefit from deeper analysis.** The paper identifies that proprietary models over-refuse in disambiguated contexts (Failure Mode 1), but does not distinguish between genuine over-refusal (model refuses to answer despite clear evidence) and legitimate caution (model correctly identifies that the question itself is bias-eliciting). Some disambiguated scenarios in BBQ involve content that safety-trained models might reasonably flag. The paper would benefit from analyzing whether over-refusal correlates with the sensitivity of the topic (e.g., higher for Religion than Gender). Given the project's current rigor and scope, this could perhaps be suggested

3. **Could benefit from analysis of false fairness.** A model that always answers "Unknown" would achieve perfect Fairness Scores on ambiguous contexts and zero on disambiguated contexts. While the harmonic mean partially addresses this, the paper does not discuss whether some models might be "trivially fair" by simply refusing everything. The Unknown Rate data in Table 6 suggests this may be happening for some proprietary models. A more explicit discussion of the fairness-capability tradeoff would be valuable.

---

### Official Review · Reviewer_zi3P · 2026-02-15

**Rating:** 4
**Confidence:** 4

**Summary:**

Authors propose a novel resource, MultiBBQ, to allow controllable fairness evaluation of MLLMs, providing two metrics to measure the fairness and the bias of models.
The contribution also engages with limitation of fairness literature, mitigating shortcuts that MLLMs tend to adopt in over-relying on textual instruction.
Authors also 1) manually evaluate the synthetic images embedded in the new resource and 2) test the effectiveness of existing mitigation methods.

**Strengths:**

Extensive experiments (28 MLLMs)

Multimodal transfer of textual only BBQ fairness benchmark

Clear, actionable findings, useful to guide current/future research (the four Fairness Failure Modes)

Even just from the Introduction, the data, method, novelty, experiments, evaluation, and findings are clear, also thanks to Figure 1 and how well written and comprehensive this first section is

**Weaknesses:**

In 2, consider adding a table summarizing the number of instances for each bias category.

It's not entirely clear/detailed how the ambiguity is introduced practically or if it was already present in the original BBQ.

The human evaluation mentioned in lines 164-165 needs further details. Who conducted it? On how many instances? W.r.t. which evaluation criteria? Which was the model that generated the images in the end?

The Evaluation Metrics paragraph, although synthetic and overall clear, could benefit from formalizing the formulas, since there's already a notation introduced and used in free-text.

Consider adding at the end of section 2 an overview of the RQs answered in the following sections, to guide the reader.

Related Work could be expanded, e.g., describing more in detail existing strategies/resources to measure fairness in multimodal settings, to properly frame the contribution.

How generalizable is the multimodal transfer for other textual only fairness datasets? This could be an interesting, useful discussion to add.

Consider adding limitations and future work to the Conclusion section.

The Appendix is extensive/huge: it's difficult to understand the relevant content / to link it to the main body. Consider adding more punctual references and possibly removing redundant tables/results.

---

### Official Review · Reviewer_otk1 · 2026-02-21
**The paper is well written and is apt to this ICLR workshop**

**Rating:** 4
**Confidence:** 4

**Summary:**

This paper focuses on profiling or bias in models and proposes a dataset to evaluate different models based on fairness and bias score. The dataset is synthetic, with images and text generated to meet the evaluation requirement. Mitigation of  The dataset contains 4 types/cases, based on visual and textual context, and then evaluate 28 models on it. Then they also perform analysis on the results to identify correlations and flag the properties that cause failures.

**Strengths:**

- Open source dataset, results, in-depth analysis of results, is very helpful in recreating the experimentation.
- A large number of models (proprietary and open-source) have been evaluated, which is helpful to gauge the trajectory of models in the area.
- This paper not only performs the evaluation, but goes on to find empirical correlations that cause failures/bias/assumptions.
- Pearsons correlation provided.

**Weaknesses:**

- Synthetic dataset is fine to begin with, but maybe capturing more real world images, stock images, would have made the dataset even stronger.
- Using more recent pictures/incidents/events from real world would have captured the accuracy of biases/assumptions that models are making.
- There is no clear distinction of the count of dataset items, which is crucial to be mentioned in the paper.

---

### Meta-Review · Area_Chair_tbAb · 2026-02-24

**Recommendation:** Main Papers Track
**Confidence:** 5

**Metareview:**

Three reviewers have evaluated the paper with overall positive votes (summary assessment: accept/accept/strong accept).

The reviews praise the contributions made by this paper. It is highlighted that the paper provides not only a new fairness evaluation benchmark for MLLMs but also provides comprehensive experimentation, along with compelling novel insights. The research and proposed benchmark is timely and of high practical relevance, and the level of detail of the experiments and their documentation is outstanding. Furthermore, the paper is clearly well aligned with the workshop theme.

There are a few pointers that could be considered for further extensions of this work, next to concrete suggestions for an expanded discussion and limitations section (including the question on how AI-generators may introduce their own biases).

Based on the overall positive reviews, I propose the paper to be accepted for the workshop.